# Dynamic control of ferroic domain patterns by thermal quenching

Jan Gerrit Horstmann [1] ✉, Ehsan Hassanpour [1], Aaron Merlin Müller [1], Yannik Zemp[1], Thomas Lottermoser [1], Yusuke Tokunaga[2], Yasujiro Taguchi [3], Yoshinori Tokura [3,4], Mads C. Weber [5] & Manfred Fiebig [1]

Controlling domain structures in ferroic materials is key to manipulating their functionality. Typically, quasi-static electric or magnetic fields are used to transform ferroic domains. In contrast, metallurgy employs rapid thermal quenches across phase transitions to create new domain patterns. This nonequilibrium approach overcomes constraints imposed by slow interactions, yet remains largely unexplored in ferroics. Here, we use thermal quenches to control ferroic domain patterns in a rare-earth orthoferrite. Cooling at variable rates triggers transitions between two ferroic phases, with transient domain evolution enabling selection of the final domain pattern. By tuning the quench rate, we either obtain the intrinsic domain structure of the low-temperature phase or transfer the high-temperature pattern–creating a hidden metastable domain state inaccessible at thermal equilibrium. Real-time imaging during quenching reveals two timescales: fast domain fragmentation followed by slower relaxation. This dynamic control of domain configurations offers a promising approach for manipulating ferroic order.

Thermal quenching of materials is ubiquitous in many areas of research and technology. In traditional sword making, for instance, rapidly cooling steel from its high-temperature austenitic state results in the transition to a metastable martensitic phase with superior mechanical properties[1,2]. More recently, thermal quenches have been applied to quantum materials, producing phases with exotic electronic characteristics[3]. As an overarching theme, these approaches aim to modify the microscopic structure of materials by rapidly changing temperature, yielding final states with novel mechanical[1,2], electrical[3–7], or magnetic properties[8–13].

Ferroics, with their variety of functional domain structures[14–16], represent particularly promising targets for quenching schemes, as evidenced by successful demonstrations of magnetic skyrmion generation[10,12,13] and the manipulation of ferro- or piezoelectric properties[17–21] through quenching. Yet, considering their broad impact on materials science, thermal quenches remain underexplored for shaping ferroic order. For instance, the domain structure of a ferroic is

intricately connected to competing long- and short-range interactions within its parent phase[15,22–24]. Phase transitions between differently ordered states involve changes in these interactions and are therefore typically accompanied by irreversible modifications of the domain pattern[22,25,26]. Changing the phase while maintaining the domain pattern would allow the transfer of mesoscopic functionality to a state with new microscopic interactions or additional coexisting orders. However, it remains to be shown if and how nonequilibrium quenches can be harnessed to control ferroic domain patterns across phase transitions between physically distinct states.

Here, we demonstrate the creation of a selected type of domain structure in the multiferroic phase of a rare-earth orthoferrite enabled by thermal quenches. Using laser illumination[8,10,27], the material is heated and rapidly cooled between two magnetically ordered phases with distinct equilibrium domain patterns. We capture the process in real time via Faraday imaging at kilohertz frame rates. Adjusting the quench rate allows switching between phases while either changing

[1]Department of Materials, ETH Zurich, Zurich, Switzerland. [2]Department of Advanced Materials Science, The University of Tokyo, Chiba, Japan. [3]RIKEN Center for Emergent Matter Science (CEMS), Saitama, Japan. [4]Department of Applied Physics and Tokyo College, The University of Tokyo, Tokyo, Japan. [5]Institut des Molécules et Matériaux du Mans, UMR 6283 CNRS, Le Mans Université, Le Mans, France. ✉e-mail: jan-gerrit.horstmann@mat.ethz.ch

the original domain pattern or preserving it. Following the real-space domain dynamics throughout the quench process reveals how spin reorientation and nonequilibrium domain evolution facilitate control over the final-state domain configuration.

## Results

### Ferroic phases and domains in DTFO

As a model system, we study single crystals of multiferroic $Dy_{0.7}Tb_{0.3}FeO_3$ (DTFO), a rare-earth orthoferrite known for its coexisting, coupled ferroic orders and its numerous magnetic phase transitions[28–30]. In DTFO, the interplay of multiple order parameters and closely spaced phase transitions within a narrow temperature range provides extensive flexibility for tuning and switching between distinct spin and charge configurations[28,30]. Below $T < T_{Neél} = 653$ K, the Fe $3d$ spins of the material arrange in a $G_xA_yF_z$ structure (in Bertaut notation, see Fig. 1a) with a dominant $G_x$-type antiferromagnetic (AFM) order (see Fig. 1a (top))[28]. Spin-canting of the Fe spins due to a Dzyaloshinskii-Moriya-type interaction induces a weak ferromagnetic (wFM) moment $F_z$ parallel to the $c$-axis of the crystal. Below a critical temperature $T_{SRT1} = 7$ K, the system undergoes a spin-reorientation transition (SRT) from a $G_xA_yF_z$ to a $F_xC_yG_z$ structure, with the wFM moment $F_x$ oriented in the sample plane along the $a$-axis (Fig. 1a (middle)). Further lowering the temperature to $T_{SRT2} < 2.5$ K drives a second SRT back to the $F_z$ structure (wFM$\|c$, Fig. 1a (bottom)). This SRT occurs concomitant with the ordering of the rare-earth moments (Dy, Tb) whose interaction with the Fe sublattice induces the additional ferroelectric (FE) order of the low-temperature (LT) multiferroic $F_z$ phase[28,30] (see phase diagram in Fig. 1c).

To demonstrate dynamic domain-pattern control, we probe the magnetic order in DTFO through its wFM component, which is accessible via linear magneto-optical imaging exploiting the Faraday effect. Figure 1b shows Faraday images of the same sample area at $T_1 > T_{SRT1}$, $T_{SRT1} > T_2 > T_{SRT2}$, and $T_3 < T_{SRT2}$. While the high-temperature (HT) $F_z$ phase shows archetypal maze structures with an average domain width of ~80 μm, the $F_x$ phase at intermediate temperature exhibits stripe domains oriented along the $a$-axis of the crystal. For

cooling under equilibrium conditions, the domain pattern of the LT-$F_z$ phase assumes a similar pattern with stripes along $a$ and additional small-scale modulations attributed to the emergent rare-earth and ferroelectric orders[28,30].

### Quench-induced domain-pattern transfer

The separation of HT- and LT-$F_z$ phases by the intermediate $F_x$ phase within a narrow in temperature range ($T_{SRT1} − T_{SRT2} < 5$ K) indicates a delicate energy landscape characterized by pronounced phase competition. This phase competition governs both the microscopic spin orientation and the mesoscopic domain structure in DTFO. Fast changes of the energy landscape may drive the system out of equilibrium, potentially providing a complementary control handle for domain shaping[3,10,31]. While rapidly varying magnetic fields are technically challenging, the combination of laser illumination with cryogenic cooling can induce positive and negative temperature changes at very high rates. Here, we investigate the domain dynamics throughout the HT-$F_z \to F_x \to$ LT-$F_z$ transition during such optically assisted thermal quenches (Fig. 1c, right).

We use a train of femtosecond laser pulses to slowly heat the sample from its LT-$F_z$ state below 2.5 K into the HT-$F_z$ phase above 7 K, followed by subsequent cooling at variable rates via time-dependent attenuation of the laser intensity. To monitor the wFM order before, during, and after the thermal quenches, we combine laser heating with high-speed Faraday imaging at a temporal resolution of $\Delta t < 500$ μs (Fig. 2a; for detailed information on the technique, see Methods Section and Fig. S1). In Faraday imaging, domain selectivity arises from the light's linear polarization plane rotating in proportion to the medium's magnetization parallel to the propagation direction of the light[32]. Selecting the clockwise or counter-clockwise rotated field component for imaging creates an intensity contrast between wFM domains with up- or down-magnetization. The sample plane, which in our case coincides with the $c$-plane of the crystal, is imaged onto an electron-multiplying charge-coupled device (EMCCD). This enables image acquisition at frame rates above 1 kHz with a high signal-to-noise ratio. The laser intensity on the sample is controlled using a half-wave plate on a fast rotation stage and a subsequent polarizer, enabling the programming of time-dependent temperature profiles, synchronized with image acquisition.

We showcase the capabilities of high-speed Faraday imaging by resolving the spin-reorientation during the HT-$F_z \to F_x$ transition (Fig. 2b) driven far from equilibrium at a quench rate of 375 K s$^{-1}$. We observe a loss of Faraday contrast between domains of opposite magnetization within 1 ms, which we attribute to the rotation of the wFM moments from their original out-of-plane to an in-plane orientation upon the HT-$F_z \to F_x$ transition[28]. The time scale of this reorientation is determined by the cooling speed across the phase transition. Snapshots of the domain pattern at different time delays across the SRT (Fig. 2b, top) show that the maze-domain pattern remains stable, while its contrast vanishes (compare residual intensities in snapshots '1.7 ms' and '2.2 ms' with initial pattern (black contours)). This suggests that during fast cooling, the orientation of the local wFM moments and the global domain configuration evolve on different time scales. Such behavior provides initial evidence that the nonequilibrium transfer of domain patterns across the SRT and into a target phase may be feasible.

To explore the potential of such quenching schemes for domain-pattern control, we investigate the wFM domain structures across both phase transitions, that is, from the HT-$F_z$ phase, passing the intermediate $F_x$ phase, and into the LT-$F_z$ phase. We image the domain patterns before and after: first, for slow cooling, allowing the domain pattern to adjust to the spin rotations (Fig. 2c); second, for a fast thermal quench driving the system far from equilibrium (Fig. 2d). Starting from maze patterns in the HT-$F_z$ phase (Fig. 2c, top), slow cooling (cooling rate $\Gamma = 0.5$ K s$^{-1}$) results in stripe patterns in the LT-$F_z$ phase (Fig. 2c, middle) with no apparent correlation between initial and final domain

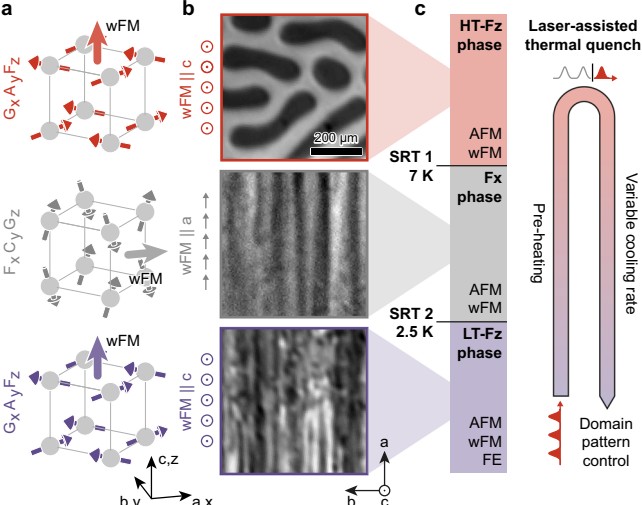

**Fig. 1 | Magnetic structure and spin reorientation transitions in DTFO.**
**a** Temperature-dependent atomic-scale magnetic structure of DTFO in Bertaut notation. Red, gray, and violet arrows, site specific spin orientation; Light red, light gray, and light violet arrows, orientation of the weak ferromagnetic moment.
**b** Faraday images of the domain pattern at $T_1 > T_{SRT1}$ (top), $T_{SRT1} > T_2 > T_{SRT2}$ (middle), and $T_3 < T_{SRT2}$ (bottom). To capture the domain image in the $F_x$ phase the sample was rotated by 45 degrees around the $b$-axis to achieve Faraday contrast.
**c** Temperature-dependent phase diagram of DTFO and laser-assisted thermal quenching scheme. Red, HT-$F_z$ phase; Gray, $F_x$ phase; Violet, LT-$F_z$ phase.

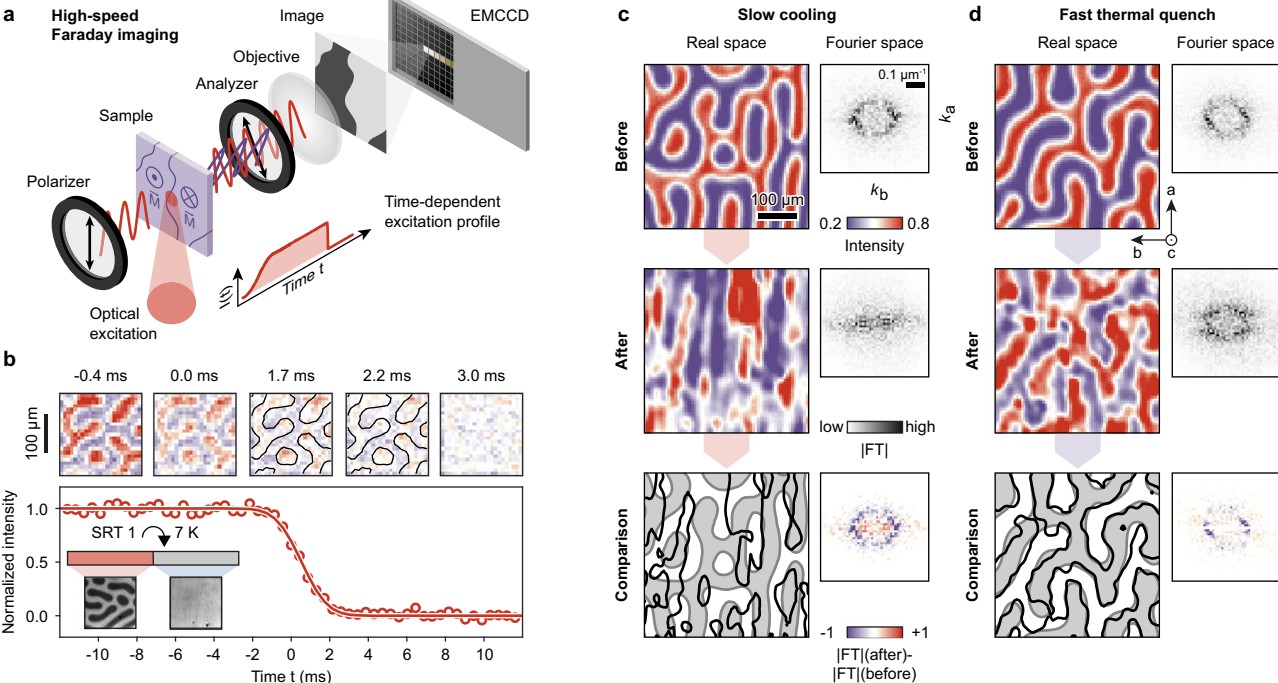

**Fig. 2 | Real-time Faraday imaging of domain dynamics and domain-pattern transfer. a** Experimental setup for time-resolved Faraday imaging at kHz frame rates. **b** Snapshots of the wFM domain evolution across SRT1. Black contours in images '1.7 ms' and '2.2 ms', show original domain pattern before the transition for comparison. **c** Domain patterns and corresponding 2D FTs recorded before (top) and after (middle) slow cooling between the HT-$F_z$ and the LT-$F_z$ phase (quench rate $\Gamma = 0.4\,\mathrm{K\,s^{-1}}$). (Bottom) Direct comparison of domain patterns recorded before and after the transition. Light gray contours, before; black contours, after. **d**, Domain patterns and corresponding 2D FTs recorded before (top) and after (middle) fast thermal quench from the HT-$F_z$ into the LT-$F_z$ phase ($\Gamma = 375\,\mathrm{K\,s^{-1}}$). (Bottom) Direct comparison of domain patterns recorded before (light gray contours) and after (black contours) the transition. The color bars shown in (**c**) also apply to the corresponding images in **d**.

patterns. The emerging anisotropy of the wFM domains is equally reflected in the vanishing of the circular feature in two-dimensional Fourier transforms (FTs) of the LT-$F_z$ pattern (compare FTs in Fig. 2c, top/middle/bottom). For fast cooling (quench rate $\Gamma = 375\,\mathrm{K\,s^{-1}}$), however, we find a markedly different behavior. Most importantly, we observe a nearly complete transfer of the initial maze pattern of the HT-$F_z$ phase to the LT-$F_z$ phase (see direct comparison of domain contours and Fourier transforms of both phases in Fig. 2d, bottom).

The above results demonstrate the ability to select between two distinct types of domain patterns, namely maze and stripe domains, in the targeted LT-$F_z$ phase. Notably, the LT maze pattern is inaccessible through an equilibrium pathway. The cooling speed represents the control parameter, which implies a critical quench rate $\Gamma_{\mathrm{crit}}$ governing the maze-domain-pattern transfer. To determine $\Gamma_{\mathrm{crit}}$, we perform a series of quench experiments at predefined quench durations $\Delta t_q$ (Fig. 3a) and analyze the similarity between initial- and final-state domain patterns (Fig. 3b; for details on the determination of the quench duration and rate, see Methods Section and Fig. S2). A comparison of images recorded before (left) and after (right) quenching highlights the gradual decrease (increase) of visual correlation for slower (faster) cooling.

We evaluate the similarity of initial and final domain structures in two metrics. The mean squared deviation (MSD) compares intensity deviations between the images in a pixel-by-pixel manner, whereas the structural similarity index (SSIM)[33] additionally accounts for correlations on larger length scales (for details, see Methods Section). Analyzing the MSD and SSIM as a function of the optical quench duration (Fig. 3c) reveals a critical range between $\Delta t_q = 200 - 1000\,\mathrm{ms}$ around a threshold value of $\Delta t_q^{\mathrm{mean}} = 600\,\mathrm{ms}$ governing the domain-pattern transfer. The latter value corresponds to a critical quench rate $\Gamma_{\mathrm{crit}} = 7.5\,\mathrm{K\,s^{-1}}$. For $\Gamma \ll \Gamma_{\mathrm{crit}}$, the LT-$F_z$ phase exhibits stripe domains. For $\Gamma \gg \Gamma_{\mathrm{crit}}$, on the other hand, the original maze-domain pattern is transferred to the LT-$F_z$ phase. We note that the quenching scheme is bidirectional: rapid temperature increases via laser heating can transfer the LT-$F_z$ pattern back into the HT-$F_z$ phase, demonstrating the reversibility of the process in principle. However, repeated cycling through the intermediate $F_x$ phase gradually imprints a stripe pattern onto the maze domains, limiting repeatability to a few cycles (see Fig. S3).

## Nonequilibrium domain evolution

We next investigate the mechanism underlying the observed dynamic domain control, focusing on how nonequilibrium domain evolution across both SRTs and the intermediate $F_x$ phase enables selecting the final domain configuration. To directly access domain dynamics in the interim $F_x$ phase, we perform time-resolved Faraday imaging on tilted samples (for details, see Fig. S4). By doing so, we achieve Faraday contrast in both $F_z$ and $F_x$ phases despite the in-plane net-magnetization of the $F_x$ phase.

We begin by examining the dynamics across SRT1 and stripe-domain formation in the $F_x$ phase during rapid quenches from $T > 7\,\mathrm{K}$ to $T = 4\,\mathrm{K}$ (Fig. 4a). For quantitative data analysis, we extract the temporal evolution of specific features in the domain pattern from FTs of the time-dependent domain configurations in a Faraday movie (Fig. 4b–d; for more information on the Fourier analysis, see Fig. S5). We find that the quench behavior splits into two processes occurring on vastly different time scales. First, SRT1 causes a rapid distortion of the long-range ordered, isotropic maze domains within tens of milliseconds (compare images recorded at '$t = 0.0\,\mathrm{ms}$' and '$t = 5.3\,\mathrm{ms}$' in Fig. 4a). In the FT, this is reflected in the fast suppression of the circular feature (compare FTs of domain patterns recorded before ('$t < 0.0\,\mathrm{ms}$') and directly after ('$t = 12-109\,\mathrm{ms}$') SRT1 in Fig. 4b, as well as the corresponding difference image). The precise evolution of the intensity of

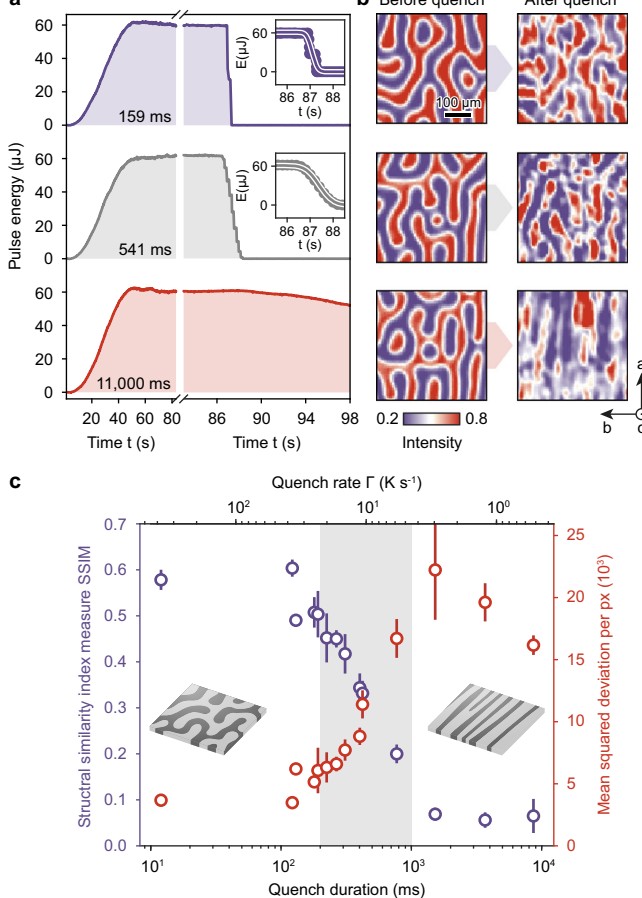

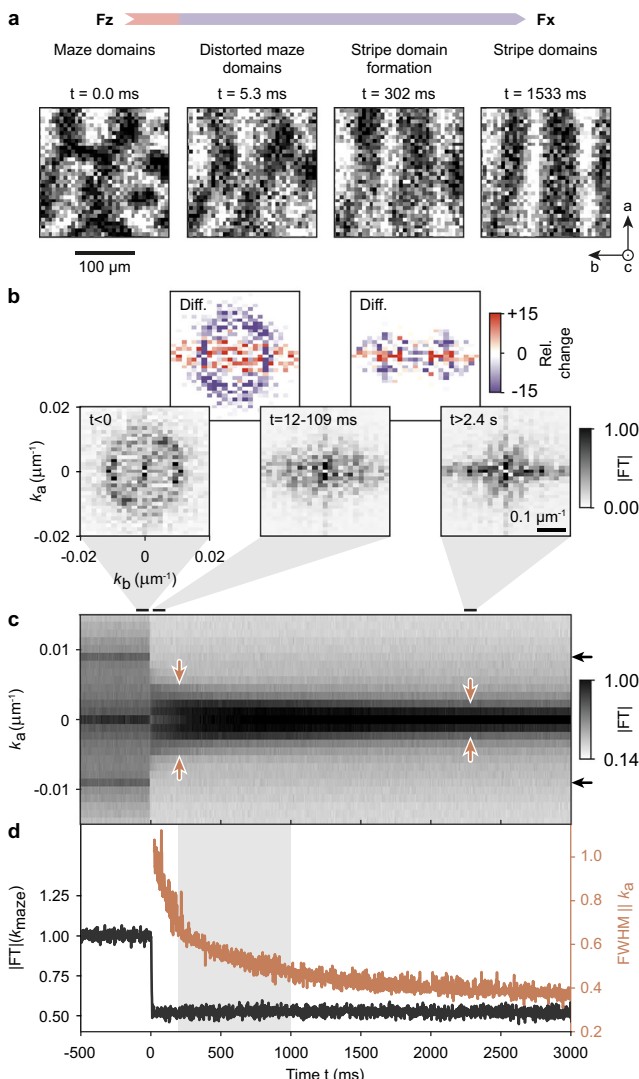

**Fig. 3 | Quench-induced magnetic domain-pattern transfer. a** Time-dependent laser-excitation profiles of three selected thermal quenches with quench durations of 159 ms (top, violet), 541 ms (middle, gray), and 11,000 ms (bottom, red). Insets: Error function fits (solid lines) to the time-dependent laser-excitation profiles (dots) in the relevant time interval. **b** Corresponding Faraday images of the same sample area recorded before (left) and after (right) the thermal quenches. Note that the images corresponding to the 11,000 ms quench (bottom) are derived from the same dataset as shown in Fig. 2c. **c** SSIM (violet circles and axis) and MSD (red circles and axis) of images recorded before and after thermal quenches as a function of the quench duration. Each data point represents the mean value of five quench measurements. Error bars, mean squared error. Light gray area, critical range of optical quench duration ($\Delta t_q$ = 200−1000 ms).

**Fig. 4 | Time-scales of spin-reorientation and domain dynamics. a** Snapshots of initial domain dynamics and stripe pattern formation in the $F_x$ phase. **b** 2D FTs of domain patterns at three selected times during the quench between the HT-$F_z$ and the $F_x$ phase (bottom) and difference images (top). Bottom row: Left, before the HT-$F_z \rightarrow F_x$ SRT ($t < 0$ ms, $|FT|_{F_z}$). Middle, directly after the SRT (12 ms $< t <$ 109 ms, $|FT|_{F_x}^{initial}$). Right, at later times ($t > 2.4$ s, $|FT|_{F_x}^{final}$). Top row: Left, difference image of 2D FTs recorded before and directly after the SRT, $(|FT|_{F_x}^{initial} - |FT|_{F_z})/\langle|FT|_{F_z}\rangle$. Right, difference image of 2D FTs recorded directly after the SRT and at later times, $(|FT|_{F_x}^{final} - |FT|_{F_x}^{initial})/\langle|FT|_{F_x}^{initial}\rangle$. **c** FTs integrated along $k_b$ as a function of time $t$ relative to SRT1. Black and brown arrows mark the features which are further analyzed in (**d**). For details on the Fourier component at $k_a = 0$ see Fig. S5. **d** Intensity of ring feature ($|FT|(k_{maze})$, black) in **b** and full width half maximum of the Fourier amplitude distribution along $k_a$ (FWHM$\|k_a$, brown) as a function of time.

this ring feature evidences that the abrupt domain-pattern change takes place on a timescale comparable to SRT1 itself (see time-dependent Fourier component $|FT|(k_{maze})$ at the spatial frequency associated with the maze pattern; black trace in Fig. 4d). The distortion of the maze pattern occurs in the form of a domain stretching (compression) along the $a$ ($b$) axis of the crystal as the evolution in Fig. 4a shows. This is also apparent from the emerging elongated feature along $k_b$ (see '$t$ = 12−109 ms' and left difference image in Fig. 4b). In other words, the well-defined maze structure breaks up into smaller stripe-like domains forming a pointillistic image of the original maze pattern. Second, the distorted maze pattern evolves into a regular stripe pattern on a longer time scale of hundreds of milliseconds (see '$t$ = 302 ms' and '$t$ = 1553 ms' in Fig. 4a). In the FT, this manifests in a narrowing of the elongated feature in $k_a$-direction over time ('$t$ = 2.4 s' in Fig. 4b, corresponding difference image, and brown trace in Fig. 4d). Importantly, the characteristic time scale of stripe-domain formation is in quantitative agreement with the critical quench time $\Delta t_q^{mean}$ = 600 ms for the domain pattern transfer into the LT-$F_z$ phase (compare light grey areas in Figs. 3c and 4d). This points to a critical

role of domain evolution in the interim $F_x$ phase in controlling the domain pattern of the targeted LT-$F_z$ phase.

To complete our picture of domain-pattern control, we now focus on the domain evolution following the SRT2 transition into the LT-$F_z$ phase below 2.5 K. To this end, we record fast real-time Faraday movies of the domain dynamics throughout the entire nonequilibrium quench process from the HT-$F_z$ through the $F_x$ to the LT-$F_z$ phase (Fig. 5a). As expected, the original maze-domain pattern of the HT-$F_z$ phase ('$t < 0$ ms') is distorted in the transition to the $F_x$ phase ('$t$ = 0−17 ms') and forms a new transient configuration ('$t$ = 30−52 ms', compare features highlighted by red circles and rectangles in first three images of Fig. 5a). More specifically, individual domains rotate towards the $a$-

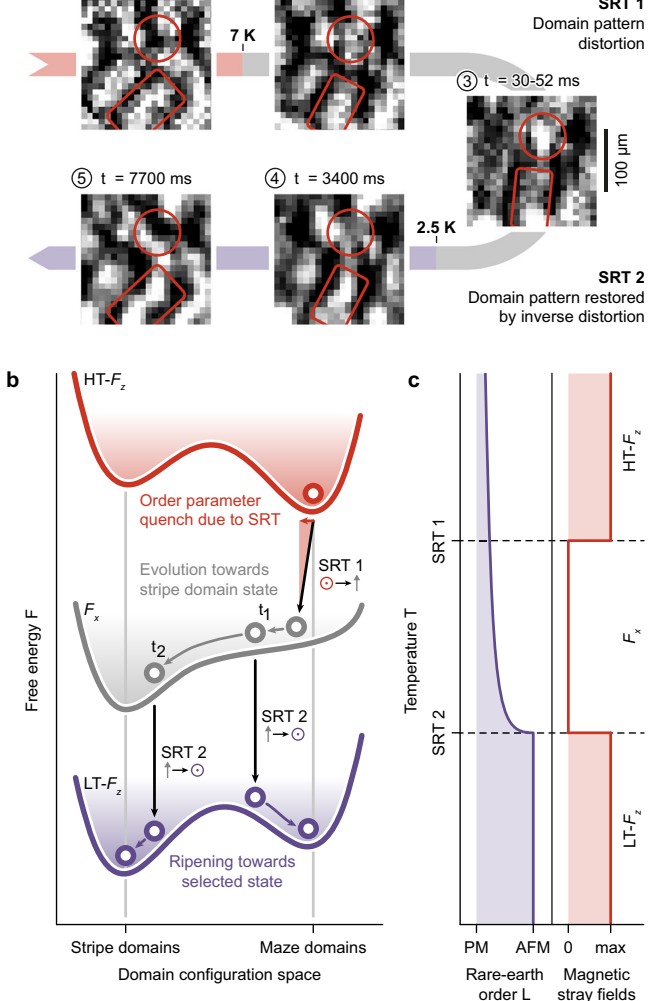

**Fig. 5 | Back-relaxation of domains and dynamic free-energy model of domain-pattern control. a** Domain evolution between HT-$F_z$ and LT-$F_z$ phases during a fast quench ($\Gamma$ = 375 K s$^{-1}$). Light red, HT-$F_z$ phase; light gray, $F_x$ phase; light violet, LT-$F_z$ phase. Red rectangles and circles highlight selected features of the domain structure discussed in the main text. **b** Dynamic free-energy model of domain-pattern control. Schematic free-energy surfaces of DTFO in domain configuration space. Red, HT-$F_z$ phase; gray, $F_x$ phase; violet, LT-$F_z$ phase. Red, gray, and violet circles represent the momentary position of the system in configuration space. **c** Illustration of the temperature-dependent contributions of rare-earth order $L$ and magnetic stray fields to the free energy $F$. PM, paramagnetic; AFM, antiferromagnetic.

axis of the crystal (see co-rotating red rectangles in (1), (2), and (3)), in line with the preferential orientation of stripe domains in the $F_x$ phase. The contraction of adjacent domains further leads to a local enhancement of the Faraday signal (highlighted by stationary red circles in (1), (2), and (3)). Remarkably, with the transition into the LT-$F_z$ phase, the distorted domains are not simply frozen in, but actively evolve back towards the original structure on a time scale of multiple seconds ((4) '$t$ = 3400 ms', (5) '$t$ = 7700 ms'). By contrast, in the adiabatic transition, this back-rotation towards the original maze structure is absent, and instead stripe domains form and ripen.

## Discussion
### Origin of magnetic domain patterns
The combined experimental observations, that is, (1) the formation of maze and stripe domains, (2) the domain-pattern transfer at high quench rates, (3) the role of domain evolution in the intermediate $F_x$ state, and (4) the back-evolution of quenched domains towards their

initial configuration in the targeted LT-$F_z$ phase can be understood within a dynamic free-energy model[34,35]. Following Baryakhtar et al.[36] and Belov et al.[37], the wFM order in rare-earth orthoferrites can be described within a Landau framework:

$$F = \int \Phi(\mathbf{M}, L_v, \nabla\mathbf{M}, \nabla L_v) - \frac{1}{2}\mathbf{M} \cdot \mathbf{H}_M. \quad (1)$$

Here, $\mathbf{M}$ denotes the local Fe magnetization, $L_v$ captures additional internal degrees of freedom, including Fe and rare-earth antiferromagnetic orders, and $\mathbf{H}_M$ represents the magnetic stray field. The generalized free energy density $\Phi$ governs internal interactions, while the term $-\frac{1}{2}\mathbf{M} \cdot \mathbf{H}_M$ stabilizes stripe and maze domains via minimization of stray fields[22].

Based on this model, Fig. 5b presents schematic sketches of the free-energy landscape in the HT-$F_z$, $F_x$, and LT-$F_z$ phases of DTFO as a function of the wFM domain configuration. In both HT-$F_z$ and LT-$F_z$ phases, the out-of-plane magnetization in a thick-film geometry gives rise to significant stray fields, favoring domain formation[22]. This is reflected in the double-well structure of the corresponding free-energy potentials in Fig. 5b, and consistent with the experimental observation of (meta-)stable maze and stripe domains in the HT-$F_z$ phase (see Figs. S3 and 1b) and the LT-$F_z$ phase (see Fig. 2c and d). In contrast, the intermediate $F_x$ phase features in-plane magnetization, eliminating surface stray fields ($\mathbf{M} \cdot \mathbf{H}_M \approx 0$) and thus the dipolar stabilization of stripe or maze domains.

Despite the absence of stray fields, stripe domains oriented along $a$ are observed when cooling slowly through the $F_x$ phase (see also the single minimum of the corresponding free energy surface in Fig. 5b). Experiments on $a$-cut samples at room temperature suggest a single-domain preference for in-plane magnetization in the HT-$F_z$ phase (see Fig. S6), indicating that the stripe pattern along $a$ observed here may arise from anisotropy effects at low temperatures. Notably, the rare-earth magnetic order in the LT-$F_z$ phase also exhibits stripe-like textures oriented along $a$[38]. We therefore propose that the stripe domains emerging in the $F_x$ phase result from a local coupling between Fe and rare-earth magnetic orders. More specifically, the anisotropy of the paramagnetically oriented rare-earth moments governs the spin reorientation of the Fe moments around $T_{SRT1}$, during which the wFM moments rotate from the $c$-axis to the $a$-axis. This reorientation implies that the effective anisotropy of the Fe moments near this temperature is of easy-plane type within the $ac$-plane. Such an anisotropy favors Bloch-type domain walls that preferentially extend within the $ac$-plane, giving rise to the stripe domain patterns aligned along the $a$-axis observed in the $F_x$ phase.

In this context, the paramagnetic orientation of rare-earth moments has been observed in DyFeO$_3$ and other rare-earth orthoferrites well above their respective long-range ordering temperatures[39–41]. The presence of such rare-earth moments oriented along $a$ is expected to introduce an anisotropy term in the free energy functional. While this term is outweighed by stray-field contributions in the HT-$F_z$ phase, it becomes dominant in the $F_x$ phase where the $\mathbf{M} \cdot \mathbf{H}_M$ term vanishes (see Fig. 5c).

The assumption of a coupling of Fe and rare-earth orders both above and below $T_{SRT2}$ is supported by two experimental observations. First, the increasing anisotropy of the HT-$F_z$ maze pattern near $T_{SRT1}$ indicates the influence of the rare-earth order even at elevated temperatures. Second, below $T_{SRT2}$, the stripe-like pointillistic distortions observed in the transferred maze pattern after fast quenches into the LT-$F_z$ phase (Fig. 3b) suggest an imprint of the rare-earth stripe order on the quenched domain structure.

### Domain dynamics during quenches
Having established equilibrium free-energy surface models to describe the HT-$F_z$, $F_x$, and LT-$F_z$ domain patterns, we now turn towards the

dynamic behavior of the system during thermal quenches across the two spin-reorientation transitions. Starting from the maze-domain configuration in the HT-$F_z$ phase, the first SRT induces a rapid initial displacement in configuration space (horizontal red arrow in Fig. 5b) and transfers the system into the $F_x$ state. For this $F_x$ phase, due to the rotation of **M** into the $ab$-plane, and thus $\mathbf{M} \cdot \mathbf{H}_M \approx 0$, the free-energy surface exhibits only one stable state, namely the stripe-domain configuration which is stabilized by the anisotropy of the paramagnetically oriented rare-earth moments. Consequently, the system evolves towards the remaining stable minimum until the second SRT restores the free-energy surface to its double-well form in the LT-$F_z$ phase. In this scenario, the time spent in the $F_x$ phase becomes a decisive parameter: For short quench times ($t_1$), the system starts to evolve towards the stripe configuration in $F_x$, but is driven back into the maze configuration on the free-energy surface of the LT-$F_z$ phase. Long quench times ($t_2$), on the other hand, facilitate the formation of stripes in the intermediate $F_x$ phase, which are then transferred to the multiferroic LT-$F_z$ phase. In both cases, gradients in the free-energy landscape force an evolution of transferred domain structures towards the local minimum in configuration space in the LT-$F_z$ phase, as observed by time-resolved Faraday imaging (compare Fig. 5a).

We note that while the quench-induced maze-domain structure represents a long-lived metastable state in the LT-$F_z$ phase (lifetime $\tau_{\mathrm{maze}}^{\mathrm{LT}-F_z} \gg 1$ h), it can only be accessed via a nonequilibrium pathway. In other words, below $T_{\mathrm{SRT2}}$, the growing anisotropy associated with the long-range-ordered rare-earth moments alters the balance of competing energy terms in Eq. (1), such that stray-field minimization is achieved through stripe rather than maze domains – making the latter inaccessible in thermal equilibrium. Consequently, the maze-domain configuration has to be imprinted onto the LT-$F_z$ phase by rapid cooling. The quench-induced maze-domain configuration remains stable throughout the temperature range of the LT-$F_z$ phase (2.0 – 2.5 K) and is robust against external magnetic fields up to $\pm 100$ mT, that is, fields comparable to that required to transform the equilibrium stripe domains into a single-domain configuration.

Finally, we consider how the final-state domain structure in the Fe and rare-earth orders impacts the ferroelectric domain pattern in the multiferroic LT-$F_z$ phase. Since the ferroelectric polarization in DTFO arises as a secondary effect – induced by exchange striction between the Fe and rare-earth sublattices – its domain configuration is not independently determined, but likely governed by the magnetic domain pattern[30]. In particular, the stripe or maze domains enforced by magnetic-anisotropy and stray-field minimization below $T_{\mathrm{SRT2}}$ are expected to dictate the FE domain morphology. Electronic defects are often known to influence ferroelectric domain patterns by locally pinning domain walls. In DTFO, however, we find no indications of such pinning in the magnetic domain morphology of the wFM order. This observation, together with the improper nature of ferroelectricity in the material, suggests that electronic defects do not play a significant role in shaping the multiferroic domain structure of the LT-$F_z$ phase. Future studies could aim to directly visualize the quench-induced ferroelectric domain pattern using second harmonic generation imaging.

Beyond the model system of DTFO, we propose that adapting thermal quenching schemes to other classes of ferroics or correlated materials with electronic phase separation could unlock access to previously unknown or hidden (domain) configurations of matter, potentially without the immediate need for ultrafast optical excitation[3]. Candidate systems range from other rare-earth orthoferrites with sharp spin-reorientation transitions[42], such as YbFeO₃[43,44], to materials with strong phase competition confined to narrow temperature windows – among them systems with electronic phase separation, including charge-density-wave compounds[3,45]. In this, fast real-time imaging[46–48] of irreversible switching events and fluctuations,

as demonstrated here, serves as an essential tool for exploring, understanding, and controlling transitions into these exotic phases.

In summary, we have demonstrated dynamic control over the domain pattern of a multiferroic phase by functionalizing phase transitions as effective switches between distinct types of domains. Our results show that fast, nonequilibrium quenches and the resulting domain dynamics not only generate new domain structures, but also enable the transfer of existing domain patterns into a target phase. Combined with external fields[49–52], all-optical control schemes[8,10,13,53–56], or the mutual coupling of different orders[30,49,57], our approach offers a powerful means for switching functional domain configurations in technologically relevant states. This refined manipulation of domains could significantly impact the development of devices that leverage microscopic ferroic heterogeneity[58].

# Methods

## Sample preparation
DTFO single crystals were grown by the floating zone method in a flow of oxygen. The crystals were cut into thin sheets of ~10 mm² expanding in the $c$-plane and lapped down to a thickness of 60 µm. This results in an optical transmission of ~10 % in the visible range. Both sample faces were polished with a silica slurry to minimize diffuse light scattering in the imaging experiments. After mechanical treatment, the crystals were annealed at temperatures $T > 700$ °C to release mechanical stress introduced by the lapping/polishing process. The samples were mounted on customized holders, the surface normal aligned either at 0 ° or at 45 ° with respect to the optical axis (see Fig. S4). For the imaging experiments, the sample holders were placed in an optical cryostat (Oxford Spectromag) and cooled to base temperatures $T_{\mathrm{base}}$ between 2 and 10 K.

## Experimental setup
In time-resolved imaging experiments, the samples were homogeneously illuminated by the collimated output of a LED (Thorlabs M660L4, central wavelength $\lambda_c = 660$ nm). The typical LED driving current in experiments ranged from 10–80 mA corresponding to optical powers of 7.6 – 56.8 µW distributed over the whole sample area. These values are vanishingly small compared with the pump-beam power. No probe-beam-induced heating was observed during measurements. A camera objective (focal length $f = 200$ mm) was used to image the sample onto the camera sensor. Between objective and sensor, the beam passes a Glan-Taylor polarizer on an adjustable rotation mount to generate the Faraday imaging contrast, and is passed through a band-pass filter (central wavelength $\lambda_c = 650$ nm, bandwidth $\Delta\lambda = 40$ nm). A schematic of the entire optical setup is shown in Fig. S1. As a detector, we employ an electron-multiplying charge coupled device (EMCCD, Andor iXon Ultra 897). In EMCCDs (Electron Multiplying Charge-Coupled Devices), electron signals from the CCD chip are amplified above the read-noise floor by a multiplication register before reaching the readout amplifier. This process enables image capture with high sensitivity at high frame rates. To achieve frame rates of up to 2350 frames per second, we use the camera's frame transfer and crop modes. An adjustable slit with four independent lamellas masks the CCD chip, leaving a pre-selected sub-area of variable size (typically 256 × 256, 128 × 128, or 64 × 64 pixel, depending on the desired spatial and temporal resolution). This setup allows to store recently recorded images in the masked sections of the chip while the current image forms in the unmasked area.

To initiate and control the wFM domain dynamics in the sample, we use the output of a femtosecond laser amplifier (Coherent Legend Elite Duo, central wavelength $\lambda_c = 800$ nm, maximum pulse energy $E_p^{\mathrm{max}} = 60 \, \mu$J, repetition rate $f_{\mathrm{rep}} = 1$ kHz, maximum average power $P_{\mathrm{cw}}^{\mathrm{max}} = 60$ mW) to heat the sample. For this purpose, the pump beam is attenuated by using a half-wave plate mounted in a fast motorized rotation stage (Thorlabs ELL14) and a subsequent, fixed Glan-Taylor

polarizer. During measurements, the temporal profile of the optical excitation, i.e., the laser attenuation time, was monitored and logged by guiding a small fraction of the pump beam onto a photodiode placed after the attenuator. The pump beam was slightly focused onto the back side of the sample by a 400-mm lens, resulting in a spot size of ~500 × 500 µm², which is large compared to the imaged sample area (for additional information, see Fig. S7). The maximum excitation fluence was 7.6 mJ cm⁻² for $P_{cw}$ = 60 mW. Synchronization between optically-assisted thermal quenches and image acquisition was realized by a customized LabView program.

## Data acquisition

For the time-resolved Faraday-imaging experiment on the SRT at $T_{SRT1}$ (Fig. 2b), the DTFO sample was heated from its initial LT-$F_x$ state at a base temperature of $T$ = 4 K into the HT-$F_z$ phase by setting increasing averaged laser power to 60 mW. The surface normal of the sample was oriented parallel to the propagation direction of the probe light, maximizing (minimizing) the domain contrast in the HT-$F_z$ ($F_x$) phase. The laser beam was rapidly attenuated by a fast mechanical shutter (shutter closing time $t_{close}$ < 3 ms) and synchronised to the EMCCD camera acquiring images (64 × 64 pixels, 2 × 2 binning, resulting image size after binning: 32 × 32 pixels) at a frame rate of 2325 Hz (corresponding temporal resolution $\Delta t_{res}$ = 430 µs). For the measurements on the quench-induced domain-pattern transfer in Fig. 2c,d and Fig. 3a,b, the sample was heated from its initial LT-$F_z$ state at a base temperature of $T$ = 2.45 K into the HT-$F_z$ phase by gradually increasing the laser heating power from 0 mW to 60 mW over a period of 30 s (see time-dependent heating trace Fig. 3a) with the attenuator described above. Following the heating step, the sample was held at maximum temperature for 10 s to stabilize the wFM domain pattern, followed by a thermal quench of predefined duration. Reference images were taken 5 s before and 5 s after the quench with typical integration times of 50 ms. To ensure the comparability between measurement runs, all steps of the experiment, that is, heating, annealing, quenching, image acquisition, and the repeated heating were fully automated. The images shown in Figs. 4a and 5a were recorded at a frame rate of 569 Hz (128 × 128 pixel) with 1 × 1 and 2 × 2 binning (in post-processing), respectively. The corresponding temporal resolution was 1.7 ms.

## Data analysis

We determine the optical quench duration $\Delta t_q$ and the quench rate $\Gamma$ through a three-step procedure. First, we measure the time-dependent laser power on the sample by recording a reference signal with a photo-diode during the laser-assisted quench (see insets in Fig. 3 for typical time-dependent laser power traces and Fig. S1 for details of the experimental setup). Fitting this signal with an error function model yields the characteristic laser attenuation time. Second, we simulate the resulting temperature dynamics using the 'heat transfer in solids' module of COMSOL® multiphysics. The laser is modeled as a Gaussian-shaped heat source whose power, following an initial equilibration period of several seconds, decays in time according to the fitted error function. As data on the heat capacity $c_p$ and heat conductivity $\kappa$ of DTFO are not available, we adopt values from the closely related compounds DyFeO₃ (DFO) and TbFeO₃ (TFO), which exhibit nearly identical thermal properties ($c_{p,TFO} \approx c_{p,DFO}$ = 10.9 J kg⁻¹ K⁻¹; $\kappa_{TFO} \approx \kappa_{DFO}$ = 10 W m⁻¹ K⁻¹; $\rho_{DFO}$ = 7.6 · 10³ kg m⁻³ [59]). The relevant optical properties of DTFO are the coefficients for absorption ($\alpha$ = 3.8 · 10⁴ m⁻¹) and reflectivity ($R$ = 0.05). The sample holder is modeled as copper with a residual resistivity ratio (RRR) of 100. For a maximum laser power of 60 mW, the simulated temperature rises from 2.4 K to 21.8 K at the laser spot center after equilibration, corresponding to temperature increase of $\Delta T$ = 19.4 K. Third, we apply the time-dependent laser power profile to extract the transient temperature distribution during the quench from the simulation. From this, we determine the times $t_{SRT1}$ and $t_{SRT2}$ at which the temperature at the laser spot center

crosses $T_{SRT1}$ = 7 K and $T_{SRT2}$ = 2.5 K, respectively. We define the quench duration as the time interval the center part of the sample spends between $T_{SRT1}$ and $T_{SRT2}$, i.e., in the $F_x$ phase. The quench rate $\Gamma$ is then defined as $\Gamma = \Delta T_{SRT}/\Delta t_q = (T_{SRT1} - T_{SRT2})/(t_{SRT2} - t_{SRT1})$. For the fastest quench (laser attenuation time: 15 ms), we find a quench duration of $\Delta t_q = t_{SRT2} - t_{SRT1}$ = 12 ms, corresponding to a maximum quench rate of $\Gamma$ = 4.5 K/12 ms = 375 K s⁻¹. This quench rate is comparable to values reported in other studies employing laser-assisted thermal quenches in correlated materials[6]. Repeating this multi-step procedure for all laser attenuation times measured in experiments yields the quench durations or rates, respectively, depicted in Fig. 3c.

To quantitatively analyze the similarity between domain patterns before and after the thermal quenches as a function of $\Delta t_q$ (see Fig. 3c), we evoke two different metrics. The mean standard deviation (MSD) or mean standard error (MSE), respectively, represents a pixel-by-pixel comparison of image intensities. For images $I^b$ and $I^a$ with $N \times M$ pixels, recorded before and after the quench,

$$\text{MSE}(I^b, I^a) = \frac{1}{NM} \sum_{i=1}^{N} \sum_{j=1}^{M} \left( I_{ij}^b - I_{ij}^a \right)^2, \quad (2)$$

where $I_{ij}^b$ ($I_{ij}^a$) is the intensity of the $ij$th pixel before (after) the quench. The structural similarity index measure (SSIM), on the other hand, compares images in terms of three different parameters: luminosity $l$, contrast $c$, and structure $s$, according to

$$\text{SSIM}(I^b, I^a) = \left[ l(I^b, I^a) \right]^\alpha \cdot \left[ c(I^b, I^a) \right]^\beta \cdot \left[ s(I^b, I^a) \right]^\gamma. \quad (3)$$

The exponents $\alpha$, $\beta$, and $\gamma$ can be chosen freely between 0 and 1 to give more weight to specific contributions. The measures of luminosity, contrast, and structure are defined by

$$l(I^b, I^a) = \frac{2\mu_b \mu_a + c_1}{\mu_b^2 + \mu_a^2 + c_1}, \quad (4)$$

$$c(I^b, I^a) = \frac{2\sigma_b \sigma_a + c_2}{\sigma_b^2 + \sigma_a^2 + c_2}, \quad (5)$$

$$s(I^b, I^a) = \frac{\sigma_b a + c_3}{\sigma_b \sigma_a + c_3}. \quad (6)$$

Here, $\mu_b$ ($\mu_a$) is the mean intensity of the image before (after) the quench, $\sigma_b^2 = 1/(NM - 1) \sum_{i=1}^{N} \sum_{j=1}^{M} (I_{ij}^b - \mu_b)^2$ is the variance of $I^b$ (analogously, $\sigma_a^2$ is the variance of $I^a$), and $\sigma_{ba} = 1/(NM - 1) \sum_{i=1}^{N} \sum_{j=1}^{M} (I_{ij}^b - \mu_b)(I_{ij}^a - \mu_a)$ the covariance of $I^b$ and $I^a$. The constants $c_1 = (k_1 L)^2$, $c_2 = (k_2 L)^2$, and $c_3 = (k_3 L)^2$ are used to stabilize the SSIM in case of small denominators. Typically, values of 0.01 and 0.03 are chosen for $k_1$ and $k_2$, respectively, whereas $L$ depends on the number of bits per pixel. In our case, $L = 2^{16} - 1$. For $\alpha = \beta = \gamma = 1$, Eq. (3) reduces to

$$\text{SSIM}(I^b, I^a) = \frac{(2\mu_b \mu_a + c_1)(2\sigma_{ba} + c_2)}{(\mu_b^2 + \mu_a^2 + c_1)(\sigma_b^2 + \sigma_a^2 + c_2)}. \quad (7)$$

Evaluating Eq. (7) as a function of the optical quench duration yields the data presented in Fig. 3b.

## Reporting summary

Further information on research design is available in the Nature Portfolio Reporting Summary linked to this article.

## Data availability

The data used in this study are available in the ETH Zurich Research Collection database under accession code 10.3929/ethz-b-000742985.

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

## Acknowledgements

This work was funded by an ETH Postdoctoral Fellowship, a Swiss National Science Foundation (SNSF) Postdoctoral Fellowship (TMPFP2 217303), and SNSF project No. 200021_215423. M.C.W. is grateful for financial support by the Région des Pays de la Loire under the Etoiles Montantes project No. 2022_11808 and the PULSAR Academy (2022_09767). The authors would like to thank P. M. Derlet, M. Müller, M. Trassin, R. V. Pisarev, and A. Vaterlaus for helpful discussions as well as J. Hecht and S. Reitz for technical support.

## Author contributions

The project was conceived by J.G.H. and M.F. Experiments and data analysis were conducted by J.G.H., with contributions from E.H., A.M.M., Y.Z., T.L. and M.C.W. Y. Tokunaga and Y. Taguchi synthesized the DTFO samples coordinated by Y. Tokura. The manuscript was written by J.G.H. All authors discussed the results and commented on the manuscript.

## Competing interests

The authors declare no competing interests.
