## [Transparent Peer Review file · Nature Communications]

Dynamic control of ferroic domain patterns by thermal quenching

Corresponding Author: Dr Jan Gerrit Horstmann

Version 0:

Reviewer comments:

Reviewer #1

(Remarks to the Author)

The article by Horstmann et al. reports on a rapid quenching process, which is able to preserve (or transform) the magnetic domain structures upon ferroic phase transformation depending on the quenching rate. The authors have used an advanced high-speed Faraday imaging technique to record the rapid transformation of the domain structures. The experimental procedure is well described and the results are convincing. However, I feel that for the article to be published in Nat. Comm., the authors should address the following points:

1. Although the material is described as multiferroic, there is no description of the ferroelectric domain structures. Could the underlying ferroelectric domain structure, and as a consequence electric defects, play a role in the transformational dynamics?
2. The authors imply the presentation of a model describing the relaxation of the domain patterns; however, only a schematic is provided in Figure 5. As far as I can tell, there is no physical model presented. Can the authors elaborate on their presented schematic and provide quantitative description, based on which some physical constants can be computed? This will lead to a more complete article, rather than just physical observations.

Reviewer #2

(Remarks to the Author)

Thermal quenching is a powerful technique for controlling phase transitions in electronic/magnetic phases that are hidden behind a competing order in condensed matter systems. In this work, the authors employ laser illumination to induce thermal quenching, enabling the selective control of ferroic domains in the multiferroic rare-earth orthoferrite $\text{Dy}_{0.7}\text{Tb}_{0.3}\text{FeO}_3$ (DTFO). The experiments are comprehensive, and the expression in both text and figures is clear. I have a few comments that need to be clarified before recommendation for publication:

1. Without quenching, the DTFO experience of spin reorientation of the Fz (HT) of maze patterns, to Fx (stripe pattern), then back to Fz (LT) of stripe pattern. While the quenching can control the domain of Fz (LT) of stripe pattern to Fz (LT) of maze patterns. What specific physical parameters are altered by the quenching process to induce these changes? The authors should provide a related discussion about the underlying mechanisms.
2. The title and introduction part talk about "the control of the ferroic domain". However, in the current works, the author's main focus is on thermal quenching control about the magnetic order/magnetic domain but lacks ferroelectric order. I am curious whether the ferroelectric state is affected when different quenching rates are applied to achieve maze or stripe domain patterns in the LT-Fz state.
3. The same question like Q2, "the control of domain pattern" is declared, while the present content of quenching technique is likely a mean of thermal method, rather than controlling the domain by purposive manipulation. So how about the repeatability of domain formation between two quenching processes?
4. a quenching rate of 300 Ks⁻¹ has been stated in the main content, how to determine such rate of temperature decreasing?
5. Some of the thermal quenching control electronic/magnetic phases are notably absent from the references list and should be included.
6. Some of the typos or descriptions should be fixed, e.g., the scalebar is missing in Fig.2b, Fig. 4a, and Fig. 5a; it's " $>2.4\text{s}$ " rather than " $>2.4\text{ms}$ " in Fig.4b, etc.

Reviewer #3

(Remarks to the Author)

The manuscript titled “Dynamic control of ferroic domain patterns by thermal quenching” by Horstmann et al. reports a investigation into the dynamic behaviors of ferroic domains under nonequilibrium conditions achieved through thermal quenching. Using high-speed Faraday imaging, the study elucidates the relationship between quenching rates and the resultant domain configurations, demonstrating both equilibrium and non-equilibrium pathways to domain evolution. A metastable domain configuration is revealed in the low-temperature LT-Fz phase in a rare-earth orthoferrite, characterized by a maze pattern that contrasts with the stripe pattern typically observed under equilibrium cooling. These findings offer valuable insights into nonequilibrium ferroic behavior. The approach developed has the potential to uncover metastable states in other systems, making this work of broad interest. I recommend the publication of this manuscript in Nature Communications after addressing the following comments.

1. The authors are encouraged to compare the thermal quench scheme presented in this study with prior advancements in the field in the introduction section.
2. Concerning the thermal quench setup shown in Fig. 2a, what is the maximum quenching rate achievable with this setup? What is the effective area influenced by the pulsed laser, and how small can the laser spot size be? Have the authors assessed the temperature distribution within the area of interest and verified its uniformity?
3. Since the cooling rate is a critical parameter in this study, how is it accurately determined? The methodology used to determine and control the cooling rate should be described in more detail. Additionally, further elaboration on the critical quenching rates and their dependence on specific material properties would enhance the reproducibility of this experiment by other researchers
4. In Fig. 1b, both the Fx and LT-Fz phases exhibit well-defined stripe domain patterns during slow cooling. However, in Fig. 2c, the stripe domain pattern (middle panel) for the slow cooling process appears more disordered. Could the authors clarify this discrepancy and provide an explanation for the apparent differences in domain pattern formation between these figures?
5. The quench durations in Fig. 3a should be clearly indicated, not just with numerical values but also with a more visual representation. Consider including a localized enlargement or graphical inset to explicitly illustrate the time width of each quench duration for better clarity.
6. For the quench duration of 11,000 ms in Fig. 3a, the post-quench image appears to be identical to the middle panel of Fig. 2c (slow cooling process). Are these images derived from the same dataset? If so, please clarify this in the manuscript; if not, could you explain the similarity between these results?
7. Have the authors investigated the stability or lifetime of the metastable phase (LT-Fz maze domain)? Does this phase spontaneously revert to the stable phase over time, and if so, under what conditions?
8. While the focus on Dy_{0.7}Tb_{0.3}FeO₃ (DTFO) is justified, a brief comparison with other potential ferroic systems would strengthen the broader applicability of the approach.
9. While the dynamic free-energy model provides valuable insights, further clarification of the underlying assumptions and potential limitations would enhance the interpretation of the results. Incorporating additional theoretical works could strengthen the analysis.

Version 1:

Reviewer comments:

Reviewer #1

(Remarks to the Author)

The authors have satisfactorily addressed the concerns raised in the earlier review. I therefore recommend the article to be published in Nature Communications

Reviewer #2

(Remarks to the Author)

The authors have thoroughly addressed the reviewers' concerns, particularly by expanding the discussion on the potential influence of quenching on ferroelectric domain configurations. Their detailed response demonstrates a solid understanding of the system. I find the manuscript substantially improved and recommend it for acceptance.

Reviewer #3

(Remarks to the Author)

I appreciate the authors' thorough revisions to the manuscript. Their responses have adequately addressed my concerns, and I therefore recommend the publication of the current version.

Dynamic control of ferroic domain patterns by thermal quenching

Jan Gerrit Horstmann^{1*}, Ehsan Hassanpour¹, Aaron Merlin Müller¹, Yannik Zemp¹, Thomas Lottermoser¹, Yusuke Tokunaga³, Yasujiro Taguchi⁴, Yoshinori Tokura^{4,5}, Mads C. Weber², Manfred Fiebig¹

¹ Department of Materials, ETH Zurich, Vladimir-Prelog-Weg 4, Zurich, 8093, Switzerland.

² Institut des Molécules et Matériaux du Mans, UMR 6283 CNRS, Le Mans Université, Le Mans, 72085, France.

³ Department of Advanced Materials Science, The University of Tokyo, Chiba, 277-8561, Japan.

⁴ RIKEN Center for Emergent Matter Science (CEMS), Saitama, 351-0198, Japan.

⁵ Department of Applied Physics and Tokyo College, The University of Tokyo, Tokyo, 113-8656, Japan.

Response to the Reviewers

We would like to sincerely thank all Reviewers for their thorough and constructive comments on both the experimental and theoretical aspects of our work. We believe their insightful feedback has significantly contributed to improving the quality and clarity of the manuscript.

In particular, all Reviewers have pointed out that the manuscript would benefit from a more in-depth discussion of the physical mechanisms underlying the quench dynamics. In response, we have developed a model based on the theoretical works of Belov *et al.*, and Bar'yakhtar *et al.* (Belov *et al.*, Soviet Physics Uspekhi **19**, 574 (1976); Bar'yakhtar *et al.*, Soviet Physics Uspekhi **31**, 810 (1988)) that accounts for the domain patterns observed in all three relevant phases by incorporating the magnetic anisotropy introduced by the paramagnetic orientation ($T > 2.5$ K) or the long-range order ($T < 2.5$ K) of the rare-earth moments. Importantly, the dynamic evolution of the system throughout thermal quenches can also be understood within this framework.

Below, we provide a detailed, point-by-point response to each of the Reviewers' comments. In the revised manuscript, changes are highlighted in blue.

Reviewer #1

1.0 Reviewer #1: „*The article by Horstmann et al. reports on a rapid quenching process, which is able to preserve (or transform) the magnetic domain structures upon ferroic phase transformation depending on the quenching rate. The authors have used an advanced high-speed Faraday imaging technique to record the rapid transformation of the domain structures. The experimental procedure is well described and the results are convincing. However, I feel that for the article to be published in Nat. Comm., the authors should address the following points:*“

Our response: We thank the Reviewer for the careful evaluation of our manuscript and for the positive comments regarding our experimental approach and the presented results. We are pleased to hear that the Reviewer found our description of the procedure clear and the findings convincing.

1.1 Reviewer #1: „*1. Although the material is described as multiferroic, there is no description of the ferroelectric domain structures. Could the underlying ferroelectric domain structure, and as a consequence electric defects, play a role in the transformational dynamics?*“

Our response: We agree that this is important point. To address it, we must consider the origin of ferroelectricity in the multiferroic phase of DTFO. As established by Tokunaga *et al.* (Tokunaga *et al.*, Nat. Mater. **8**, 558-562 (2009); Tokunaga *et al.*, Nat. Phys. **8**, 838-844 (2012)) and earlier work from our group (Hassanpour *et al.*, Science **377**, 1109-1112 (2022)), the dominant orders in DTFO (just like in the related orthoferrites DyFeO₃ and GdFeO₃) are the G_x-type Fe and G_x-type rare-earth orders. The ferroelectric polarization in these materials arises as a secondary effect via exchange striction between the Fe on the one hand, and the Dy/Tb/Gd spins on the other hand. In other words, ferroelectricity in DTFO is of an improper nature. Consequently, the ferroelectric domain structure is not independently defined but follows the magnetic domain configuration (see also Hassanpour *et al.*, Science **377**, 1109-1112 (2022)). Specifically, the stripe or maze domains stabilized by magnetic anisotropy and stray-field minimization below T_{SRT2} are expected to directly determine the ferroelectric domain morphology.

As for the potential impact of electric defects—i.e., structural or chemical imperfections that affect the local electric field, polarization, or charge distribution—on the ferroelectric domain pattern in DTFO, we do not observe any experimental signatures indicative of such defects. Typically, the presence of electric defects leads to local pinning of the ferroelectric domain structure. In this context, the mutual coupling between the ferroelectric order and the AFM/wFM orders of the Fe and rare-earth ions implies that any defect-induced pinning should manifest itself in our observables—namely, the wFM domain pattern. However, apart from the reported quench-induced transfer of the wFM domain pattern, no indication of domain pinning is observed in any of the material's phases.

Moreover, we find that the multiferroic domain pattern formed in the LT- F_z phase varies from one cooldown to the next, indicating that the domain formation process is inherently stochastic. In several of our previous studies, we investigated the characteristic domain configurations not only as a function of temperature but also under applied magnetic and electric fields. No indication of domain pinning was observed in any of those experiments.

In summary, we believe that the secondary nature of ferroelectricity in DTFO, together with the absence of any observable pinning of the multiferroic domain pattern in our experiments, strongly suggests that electric defects do not govern the observed transformation behavior.

In the revised manuscript, we have added a paragraph to the discussion section that addresses the secondary character of the ferroelectric order in the multiferroic phase of DTFO, the absence of pinning effects, and their implications for the quench-induced ferroelectric domain pattern.

1.2 Reviewer #1: „2. *The authors imply the presentation of a model describing the relaxation of the domain patterns; however, only a schematic is provided in Figure 5. As far as I can tell, there is no physical model presented. Can the authors elaborate on their presented schematic and provided quantitative description, based on which some physical constants can be computed? This will lead to a more complete article, rather than just physical observations.*“

Our response: We agree with the Reviewer that a more meaningful and accurate theoretical description of our experimental observations would make the manuscript more complete and contribute to a better understanding of the quench dynamics. In a substantial revision of the discussion section, we have therefore expanded our description of the processes underlying the observed equilibrium domain configurations, as well as the dynamic quenches and domain formation by incorporating a theoretical perspective based on Landau theory. To this end, we adapt the theoretical frameworks developed by Baryakhtar *et al.* (Baryakhtar *et al.*, Usp. Fiz. Nauk. **156**, 47-92 (1988)) and Belov *et al.* (Belov *et al.*, Sov. Phys. Usp. **19**, 574 (1976)), which were formulated to describe magnetic phenomena in rare-earth orthoferrites. Within this Landau model, we discuss the impact of stray fields and the paramagnetically oriented (HT- F_z , F_x) and

long-range-ordered (LT- F_z) rare-earth moments on the wFM domain patterns in the three relevant phases.

As our main result, we find that the observed (meta-)stable domain patterns in the HT- F_z , F_x , and LT- F_z phases are governed by the balance between the stray-field term $-1/2(\mathbf{M}\cdot\mathbf{H}_M)$ in the Landau free-energy functional and an anisotropy term originating from the preferential orientation of rare-earth moments along the a -direction of the crystal. Whereas the impact of stray fields on magnetic domain patterns in thick-film samples is well understood in experiment and theory (Hubert and Schäfer, *Magnetic Domains*, DOI: 10.1007/978-3-540-85054-0 (1998)), the paramagnetic orientation of rare-earth moments above the nominal ordering temperature in orthoferrites is, perhaps, less known (Vilarinho *et al.*, *Sci. Rep.* **12**, 9697 (2022); Zhao *et al.*, *PRB* **93**, 014417 (2016); Staub *et al.*, *PRB* **96**, 174408 (2017)). However, it is, in fact, the increasing preferential orientation of the rare-earth moments with decreasing temperature that drives the spin-reorientation transitions in this class of materials. We have recently investigated the domain pattern of the rare-earth order in the multiferroic LT- F_z phase of DTFO in a separate publication (Zemp *et al.*, arXiv:2505.16085). In the case of field cooling—i.e., when a single-domain wFM is realized—the rare-earth order forms a stripe domain pattern along the a -direction, highlighting the preferential orientation of the rare-earth moments along this a -direction and supporting our model.

Based on these considerations, we propose the following model: At high temperatures ($T \gg 10$ K) where the rare-earth moments are randomly oriented, the domain configuration is primarily governed by stray-field minimization. This leads to the formation of isotropic maze domains. As the temperature decreases to around 10 K, the rare-earth moments begin to exhibit a preferential orientation along a (but not yet long-range ordering), which introduces an anisotropy term into the system. In our experiments, this manifests as a slight preferential alignment of maze domains along the a -direction, particularly near the first spin reorientation transition (SRT1) at 7 K. At SRT1, the Fe spins rotate into the ab -plane, causing the stray-field contribution ($\mathbf{M}\cdot\mathbf{H}_M$) to vanish. As a result, the anisotropy associated with the paramagnetically oriented rare-earth moments becomes dominant and enforces a stripe-like domain configuration of the weak ferromagnetic (wFM) order, even in the absence of stray fields. After SRT2, the wFM moments reorient along the c -axis, i.e., perpendicular to the sample surface, marking the onset of the multiferroic LT- F_z phase. Here, the domain pattern reflects a competition between stray-field minimization and the anisotropy along a , now due to the long-range order of the rare-earth moments below 2.5 K. Thus, in thermal equilibrium, stray fields in this phase are minimized by stripes instead of maze domains. Within this framework, the LT- F_z maze structure appears as a metastable state that can only be realized through rapid thermal quenches. Following such a quench, the system locally attempts to accommodate the anisotropy imposed by the rare-earth order, resulting in the emergence of micron-scale stripe segments aligned along the a -direction. Experimentally, we observe these as characteristic “pointillistic” distortions within the quenched maze domains.

While this phenomenological model of the domain dynamics in terms of Landau theory already offers several valuable insights, we agree with the Reviewer that a quantitative description, e.g. in terms of numerical simulations would be desirable. However, a fully quantitative description in terms of numerical simulations is beyond the scope of the present work. While such simulations could, in principle, reproduce aspects of the domain configurations, we believe that they would not yield substantially deeper physical insights without detailed knowledge of various material parameters (e.g., exchange constants, anisotropies, domain wall energies, and temperature-dependent rare-earth order parameters), many of which are not available for DTFO at present. As is often the case with such simulations, the predictive power is ultimately limited by the quality and completeness of the input parameters—one often only retrieves what has been put in. In this context, we consider our phenomenological, Landau-based approach sufficient to capture the

key physics governing the observed domain evolution and to motivate the central findings of our study.

In the revised manuscript, we have added an extensive discussion of the above points, including the Landau framework, the role of magnetic stray fields, and the impact of the partially and long-range-ordered rare-earth moments.

Reviewer #2

2.0 Reviewer #2: *„Thermal quenching is a powerful technique for controlling phase transitions in electronic/magnetic phases that are hidden behind a competing order in condensed matter systems. In this work, the authors employ laser illumination to induce thermal quenching, enabling the selective control of ferroic domains in the multiferroic rare-earth orthoferrite Dy_{0.7}Tb_{0.3}FeO₃(DTFO). The experiments are comprehensive, and the expression in both text and figures is clear. I have a few comments that need to be clarified before recommendation for publication.“*

Our response: We thank the Reviewer for the thorough review of our manuscript and for the positive feedback on our experimental work, as well as the clarity of both the text and figures. We appreciate the encouraging comments and address the specific points raised below.

2.1 Reviewer #2: *1. Without quenching, the DTFO experience of spin reorientation of the F_z (HT) of maze patterns, to F_x (stripe pattern), then back to F_z (LT) of stripe pattern. While the quenching can control the domain of F_z (LT) of stripe pattern to F_z (LT) of maze patterns. What specific physical parameters are altered by the quenching process to induce these changes? The authors should provide a related discussion about the underlying mechanisms.*

Our response: This is an important point, which was also highlighted, in essence, by Reviewer #1. To discuss the impact of specific physical parameters, we evoke a Landau theory (for details, please see our response to question number two of Reviewer #1). Summarizing our findings, we identify two main physical parameters whose balance impacts the domain patterns and their dynamics in all three relevant phases and thus facilitate dynamic control over the LT-F_z domain configuration: (1) Magnetic stray fields in the HT-F_z and LT-F_z phases as well as their absence in the F_x phase due to the spin reorientation in between SRT1 and SRT2; (2) an anisotropy in the system along the *a*-direction of the crystal due to the preferential orientation (long-range ordering) of the rare-earth moments above (below) T = 2.5 K.

Based on these two parameters, we argue that rapid quenching effectively bypasses the transformation from maze-like to stripe-like domains that would normally occur in the F_x phase. Within the framework of Landau theory, we argue that this transformation is driven by the rotation of the spins into the *ab*-plane at the first spin reorientation transition (SRT1). This reorientation eliminates the stray-field contribution to the free energy, thereby shifting the energetic balance in favor of the rare-earth-induced anisotropy term, which promotes the formation of stripe domains. In this context, the critical quench time observed in our experiments reflects the characteristic timescale for the maze-to-stripe domain transformation, following a quasi-instantaneous change in the free energy landscape induced by spin reorientation.

In the revised manuscript, we have added an extended discussion of the mechanisms underlying the domain dynamics and domain pattern control, i.e. magnetic stray fields and the paramagnetic orientation/long-range ordering of rare-earth moments.

2.2 Reviewer #2: 2. *The title and introduction part talk about "the control of the ferroic domain". However, in the current works, the author's main focus is on thermal quenching control about the magnetic order/magnetic domain but lacks ferroelectric order. I am curious whether the ferroelectric state is affected when different quenching rates are applied to achieve maze or stripe domain patterns in the LT-F_z state.*

Our response: We agree with the Reviewer that this is an important point to discuss. A similar concern was also raised by Reviewer #1, and we have provided a detailed response to this in our reply to Question 1 from Reviewer #1. We kindly refer Reviewer #2 to that discussion for a comprehensive explanation.

In response to the important concerns raised by Reviewers #1 and #2, we have amended the manuscript by a paragraph discussing the potential impact of quenches on the ferroelectric domain pattern of the material.

2.3 Reviewer #2: 3. *The same question like Q2, "the control of domain pattern" is declared, while the present content of quenching technique is likely a mean of thermal method, rather than controlling the domain by purposive manipulation. So how about the repeatability of domain formation between two quenching processes?*

Our response: For a single down-up-down quench cycle, we observe a strong correlation between the resulting domain patterns, indicating a certain degree of repeatability. However, when a larger number of successive quenches are applied, repeated transitions through the F_x phase result in the progressive imposition of a stripe-like domain configuration onto the initially maze-like pattern. This cumulative effect limits the reproducibility of the original domain configuration to only a few cycles. While domain fluctuations and ripening in the equilibrium HT- F_z and LT- F_z phases certainly introduce a statistical element following each quench, we believe that the dominant factor limiting the reproducibility of a given domain pattern after multiple cycles is the repeated traversal of the F_x phase, in which stripe domains represent the only stable configuration.

Moreover, if one starts from two distinct wFM domain configurations in the HT- F_z phase, the resulting LT- F_z domain patterns obtained after rapid thermal quenching will generally remain distinguishable. That is, quenching does not cause convergence toward a universal domain configuration but rather retains some memory of the initial state — provided the number of quenches remains low.

In the manuscript, we have added a statement on the repeatability of the domain pattern transfer that discusses the impact of repeatedly traversing the F_x phase.

2.4 Reviewer #2: 4. *a quenching rate of 300 Ks⁻¹ has been stated in the main content, how to determine such rate of temperature decreasing?*

Our response: In the initial submission, the quench rate was defined as the temperature difference between SRT1 and SRT2 (7 K – 2.5 K = 4.5 K) divided by the laser attenuation time, which was measured for each individual quench by a reference photodiode. For the fastest quench, corresponding to the shortest laser attenuation time (15 ms), this yielded a maximum quench rate of 4.5 K / 15 ms = 300 K s⁻¹. However, this estimate assumed that (1) the sample temperature within the probed area instantaneously follows the time-dependent laser heating, (2) the temperature decrease is linear, and (3) the time spent above T(SRT1) during the quench can be neglected.

In response to the Reviewer's concerns, we have revisited and refined our analysis. Specifically, we start from the experimentally measured laser attenuation time, determined by fitting an error function to the time-dependent signal recorded by the reference photodiode, as described in the Methods section. This procedure provides the laser power on the sample as a function of time. Using the measured laser power and the known laser spot diameter as inputs, we then simulate the spatially and temporally resolved temperature evolution on the sample by performing COMSOL simulations within the "Heat Transfer in Solids" module. The laser spot is modeled as a Gaussian-shaped temperature profile, and the laser power is reduced according to a time-dependent error function with a width matching the measured laser attenuation time (see Fig. S5 and the Methods section for material parameters and further details).

The simulations yield two main results: First, they confirm that the sample temperature tracks the laser heating quasi-instantaneously on the millisecond timescale relevant for our experiments. This conclusion is consistent with an independent estimate based on the sample's heat capacity and thermal conductivity (using DyFeO₃ parameters due to the lack of specific data for DTFO; we could have chosen TbFeO₃ as well, since DyFeO₃ and TbFeO₃ have nearly identical thermodynamic properties). Second, by following the time evolution of the temperature at the center of the laser spot, we can define the quench duration. We now define the quench duration as the time the system spends between the two spin reorientation transitions, i.e., in the F_x phase. More precisely, in the simulations we evaluate the times $t(\text{SRT1})$ and $t(\text{SRT2})$ at which the temperature crosses $T(\text{SRT1}) = 7 \text{ K}$ and $T(\text{SRT2}) = 2.5 \text{ K}$, respectively. The (maximum) quench rate is then given by the temperature difference divided by the time difference: $\text{quench rate} = (T(\text{SRT1}) - T(\text{SRT2})) / |t(\text{SRT2}) - t(\text{SRT1})|$. For the fastest quench, with a laser attenuation time of 15 ms, we find $|t(\text{SRT2}) - t(\text{SRT1})| = 12 \text{ ms}$, leading to a maximum quench rate of $4.5 \text{ K} / 12 \text{ ms} = 375 \text{ K s}^{-1}$.

In the revised manuscript, we have updated the Methods section with a detailed description of the quench rate determination procedure and have added Fig. S5 to the Supplementary Information to aid understanding. We have also cross-referenced the Methods section more prominently in the main text.

2.5 Reviewer #2: *5. Some of the thermal quenching control electronic/magnetic phases are notably absent from the references list and should be included.*

Our response: We would like to emphasize that the absence of certain references was not intentional but reflected our knowledge of the field at the time of writing. In response to the Reviewer's suggestion, we have conducted an extensive additional literature review and expanded the list of cited works accordingly (see Refs. 3-13 and 17-21). Should we have overlooked any further important contributions, we would be grateful to include them in the revised manuscript.

2.6 Reviewer #2: *6. Some of the typos or descriptions should be fixed, e.g., the scalebar is missing in Fig.2b, Fig. 4a, and Fig. 5a; it's " $t > 2.4 \text{ s}$ " rather than " $t > 2.4 \text{ ms}$ " in Fig.4b, etc.*

Our response: We thank the Reviewer for bringing these points to our attention; we have addressed them accordingly in the revised manuscript. Once again, we are grateful for the Reviewer's valuable comments and hope that he/she finds the revised version improved.

Reviewer #3

3.0 Reviewer #3: *The manuscript titled "Dynamic control of ferroic domain patterns by thermal quenching" by Horstmann et al. reports a investigation into the dynamic behaviors of ferroic domains under nonequilibrium conditions achieved through thermal quenching. Using high-*

speed Faraday imaging, the study elucidates the relationship between quenching rates and the resultant domain configurations, demonstrating both equilibrium and non-equilibrium pathways to domain evolution. A metastable domain configuration is revealed in the low-temperature LT-Fz phase in a rare-earth orthoferrite, characterized by a maze pattern that contrasts with the stripe pattern typically observed under equilibrium cooling. These findings offer valuable insights into nonequilibrium ferroic behavior. The approach developed has the potential to uncover metastable states in other systems, making this work of broad interest. I recommend the publication of this manuscript in Nature Communications after addressing the following comments.

Our response: We thank the Reviewer for the very positive and constructive feedback on our manuscript, and for recognizing the broader relevance and potential impact of our work.

3.1 Reviewer #3: *1. The authors are encouraged to compare the thermal quench scheme presented in this study with prior advancements in the field in the introduction section.*

Our response: We agree with the Reviewer that the manuscript would benefit from a comparison of our approach with other thermal quenching schemes. In the revised manuscript, we have amended the introduction by a corresponding statement that highlights previous contributions to the field, including magnetic skyrmion generation (Büttner *et al.*, Nat. Mater. **20**, 30–37 (2021); Oike *et al.*, Nat. Phys. **12**, 62–66 (2016); Berruto *et al.*, PRL **120**, 117201 (2018)) as well as the manipulation of ferro- or piezoelectric properties (Lee *et al.*, ACS Appl. Electron. Mater. **1**, 1772–1780 (2019); Muramatsu *et al.*, JJAP **55**, 10TB07 (2016); Poudel Chhetri *et al.*, J. Appl. Phys. **132**, 045107 (2022); Bai *et al.*, J. Adv. Ceram. **9**, 511–516 (2020); Zhang *et al.*, APL **87**, 262907 (2005)) through quenching (see also Refs. 3-13 and 17-21). Should we have overlooked other important works in the field, we would be grateful to include them in the revised manuscript.

3.2 Reviewer #3: *2. Concerning the thermal quench setup shown in Fig. 2a, what is the maximum quenching rate achievable with this setup? What is the effective area influenced by the pulsed laser, and how small can the laser spot size be? Have the authors assessed the temperature distribution within the area of interest and verified its uniformity?*

Our response: We thank the Reviewer for bringing up these important technical aspects. In our experiments, the maximum achievable quenching rate of 375 Ks^{-1} is primarily limited by the thermal conductivity of both the sample and the sample holder, which govern the efficiency of heat extraction following laser excitation (see also our response to question 4 of Reviewer #2). To estimate the area effectively influenced by the pulsed laser, we recorded full-field images of the sample under laser illumination (see, e.g., Fig. S6a). The laser intensity on the sample surface follows a Gaussian profile, resulting in a corresponding Gaussian-shaped temperature distribution. We use the characteristic domain patterns and contrast variations associated with different phases to deduce the local equilibrium temperature distribution, allowing us to identify the region significantly affected by the optical excitation. Furthermore, we can directly image the pump laser spot (pump wavelength $\lambda = 800 \text{ nm}$) on the sample with our camera. To this end, we remove the narrowband filter used to block the pump and transmit the 650 nm probe light from the LED. In all experiments presented in this work, we employed a circular laser focal spot with a diameter of $500 \mu\text{m}$ FWHM. We ensured that the area of interest remained well within this excitation region to minimize the influence of temperature gradients at the spot's periphery. Finally, in experiments involving very localized optical excitation, we observe that the temperature or phase front propagates through the material at a speed of approximately $10^5 \mu\text{m/s}$ (see Fig. S6c). Consequently, we expect the temperature distribution to equilibrate across the entire sample within a few milliseconds — a timescale that is fast compared to the domain dynamics investigated at larger length scales in this work.

Regarding the second part of the Reviewers question, in principle, the laser focus can be reduced to a diameter of about 10 μm , which is limited by the optical geometry—specifically, the minimum working distance between the cryostat center and the focusing lens located outside the cryostat.

Concerning the final question, our COMSOL simulations allow us to extract the spatially dependent temperature distribution across the imaged area of the sample (see Fig. S6b). We find that the maximum relative temperature difference, i.e., between the center and the edge of the area of interest, amounts to less than 10%.

In the revised manuscript, we have added clarifying statements to the Methods Section and amended Fig. S6 to show how the laser focal spot, temperature gradients on the sample, and the propagation of the temperature front can be characterized.

3.3 Reviewer #3: *3. Since the cooling rate is a critical parameter in this study, how is it accurately determined? The methodology used to determine and control the cooling rate should be described in more detail. Additionally, further elaboration on the critical quenching rates and their dependence on specific material properties would enhance the reproducibility of this experiment by other researchers.*

Our response: We thank the Reviewer for bringing up this point. A similar concern was raised by Reviewer #2 (question 4), and we kindly refer Reviewer #2 to that discussion for detailed explanations.

3.4 Reviewer #3: *4. In Fig. 1b, both the F_x and $LT-F_z$ phases exhibit well-defined stripe domain patterns during slow cooling. However, in Fig. 2c, the stripe domain pattern (middle panel) for the slow cooling process appears more disordered. Could the authors clarify this discrepancy and provide an explanation for the apparent differences in domain pattern formation between these figures?*

Our response: While there is indeed some statistical distribution in the domain sizes of the wFM order in the $LT-F_z$ phase, the mean domain size remains relatively constant. An extended comparison reveals that the perceived increased distortion in Fig. 2c (middle panel) is caused by differences in image scaling rather than by an actual change in domain structure. Specifically, the scales of the two images differ roughly by a factor of two (as can be verified by comparing the respective scale bars). Combined with the use of different color schemes, this difference may have created the impression of a stronger distortion in Fig. 2c (middle panel).

3.5 Reviewer #3: *5. The quench durations in Fig. 3a should be clearly indicated, not just with numerical values but also with a more visual representation. Consider including a localized enlargement or graphical inset to explicitly illustrate the time width of each quench duration for better clarity.*

Our response: We agree with the Reviewer and have added graphical insets to Fig. 3a to illustrate the time scale of each quench and, in addition, show the error function model fitted to the data in order to determine the laser attenuation time.

3.6 Reviewer #3: *6. For the quench duration of 11,000 ms in Fig. 3a, the post-quench image appears to be identical to the middle panel of Fig. 2c (slow cooling process). Are these images derived from the same dataset? If so, please clarify this in the manuscript; if not, could you explain the similarity between these results?*

Our response: We thank the Reviewer for pointing this out. Indeed, both figures are derived from the same dataset. We have clarified this point in the revised manuscript by adding a statement to the caption of Fig. 3.

3.7 Reviewer #3: *7. Have the authors investigated the stability or lifetime of the metastable phase (LT- F_z maze domain)? Does this phase spontaneously revert to the stable phase over time, and if so, under what conditions?*

Our response: The lifetime of the metastable LT- F_z maze domain configuration is indeed a highly interesting parameter, as its measurement would allow for an estimate of the energy barrier separating the maze and stripe domain configurations. Within the scope of this work, we have monitored the quench-induced domain structure for up to one hour following a single thermal quench and found no indication of a spontaneous reversion of the domain pattern toward the stripe configuration. The maze pattern can be erased either by applying a static magnetic field ($B > 100$ mT) along the c-axis or by thermally cycling into adjacent phases of the phase diagram. The magnetic field required to reorient the wFM moments in the metastable maze-like LT- F_z phase—and thereby create a single-domain state in M—is comparable to the field needed in the stripe-like LT- F_z phase.

In the revised manuscript, we have included a statement on the stability of the quench-induced domain configuration in the LT- F_z phase, its longevity, as well as susceptibility to external magnetic fields.

3.8 Reviewer #3: *8. While the focus on Dy_{0.7}Tb_{0.3}FeO₃ (DTFO) is justified, a brief comparison with other potential ferroic systems would strengthen the broader applicability of the approach.*

Our response: We thank the Reviewer for this suggestion. Indeed, we believe that the transfer of rapid thermal quenches from DTFO to other types of ferroics or correlated materials in general could facilitate enhanced control over and provide access to functional (metastable) states. We have therefore amended the discussion section of the manuscript by adding a paragraph on the potential transfer of laser-induced quenching schemes to other rare-earth orthoferrites, as well as to other classes of materials exhibiting sharp phase transitions and/or significant (ferroic) phase competition.

3.9 Reviewer #3: *9. While the dynamic free-energy model provides valuable insights, further clarification of the underlying assumptions and potential limitations would enhance the interpretation of the results. Incorporating additional theoretical works could strengthen the analysis.*

Our response: We agree that the manuscript would profit considerably from additional theoretical works, as pointed out by all Reviewers. We have therefore devised a theoretical framework based on Landau theory to support our dynamic free energy model (see also our response to Reviewers #1 and #2). We believe that the amended theoretical discussion of our experimental observations strengthens our analysis considerably and thank the Reviewer yet again for his/her suggestions in this direction.

We would like to thank all Reviewers once again for their helpful comments and hope that they find the revised manuscript improved.

Dynamic control of ferroic domain patterns by thermal quenching

Jan Gerrit Horstmann^{1*}, Ehsan Hassanpour¹, Aaron Merlin Müller¹, Yannik Zemp¹, Thomas Lottermoser¹, Yusuke Tokunaga³, Yasujiro Taguchi⁴, Yoshinori Tokura^{4,5}, Mads C. Weber², Manfred Fiebig¹

¹ Department of Materials, ETH Zurich, Vladimir-Prelog-Weg 4, Zurich, 8093, Switzerland.

² Institut des Molécules et Matériaux du Mans, UMR 6283 CNRS, Le Mans Université, Le Mans, 72085, France.

³ Department of Advanced Materials Science, The University of Tokyo, Chiba, 277-8561, Japan.

⁴ RIKEN Center for Emergent Matter Science (CEMS), Saitama, 351-0198, Japan.

⁵ Department of Applied Physics and Tokyo College, The University of Tokyo, Tokyo, 113-8656, Japan.

Response to the Reviewers (round 2)

We would like to thank all Reviewers for their careful reevaluation of our revised manuscript. We reiterate that their insightful feedback has greatly contributed to improving the quality of the work. Below, we provide a point-by-point response to each of the Reviewers' comments.

Reviewer #1

1.0 Reviewer #1: „*The authors have satisfactorily addressed the concerns raised in the earlier review. I therefore recommend the article to be published in Nature Communications.*“

Our response: We thank the Reviewer for their valuable input throughout the review process and appreciate the timely evaluation and recommendation for publication.

2.0 Reviewer #2: „*The authors have thoroughly addressed the reviewers' concerns, particularly by expanding the discussion on the potential influence of quenching on ferroelectric domain configurations. Their detailed response demonstrates a solid understanding of the system. I find the manuscript substantially improved and recommend it for acceptance.*“

Our response: We thank the Reviewer for the positive evaluation and their thoughtful comments throughout the review process. We are pleased that the expanded discussion on the influence of quenching on ferroelectric domain configurations was helpful, and we appreciate the recognition of the improvements made to the manuscript.

3.0 Reviewer #3: „*I appreciate the authors' thorough revisions to the manuscript. Their responses have adequately addressed my concerns, and I therefore recommend the publication of the current version.*“

Our response: We sincerely thank the Reviewer for their constructive feedback and supportive recommendation. We are glad that the revisions addressed the raised concerns and appreciate the positive assessment of the revised manuscript.